# Molecular Docking of Isolated Alkaloids for Possible α-Glucosidase Inhibition

**DOI:** 10.3390/biom9100544

**Published:** 2019-09-27

**Authors:** Noor Rahman, Ijaz Muhammad, Haroon Khan, Michael Aschner, Rosanna Filosa, Maria Daglia

**Affiliations:** 1Department of Biochemistry, Abdul Wali Khan University Mardan, Mardan-23200, KP, Pakistan; noorbiochemist@gmail.com; 2Department of Zoology, Abdul Wali Khan University Mardan, Mardan-23200, KP, Pakistan; ijazmawkum@gmail.com (I.M.); nayabawkum@gmail.com (G.-E.-N.); 3Department of Pharmacy, Abdul Wali Khan University Mardan, Mardan-23200, KP, Pakistan; 4Department of Molecular Pharmacology, Albert Einstein College of Medicine Forchheimer 209 1300 Morris Park Avenue Bronx, NY 10461, USA; michael.aschner@einstein.yu.edu; 5Department of Experimental Medicine, University of Campania “Luigi Vanvitelli”, via L. De Crecchio 7, 80138 Naples, Italy; Rosanna.FILOSA@unicampania.it; 6Consorzio Sannio Tech-AMP Biotec, Appia Str. 7, 82030 Apollosa, BN, Italy; 7Department of Pharmacy, University of Naples Federico II, 80138 Naples, Italy; 8International Research Center for Food Nutrition and Safety, Jiangsu University, Zhenjiang 212013, China

**Keywords:** α-glucosidase, plant alkaloids, molecular docking, new drug discovery

## Abstract

Diabetes mellitus, one of the most common endocrine-metabolic disorders, has caused significant morbidity and mortality worldwide. To avoid sugar digestion and postprandial hyperglycemia, it is necessary to inhibit α-glucosidase, a digestive enzyme with an important role in carbohydrate digestion. The criteria for the selection of alkaloids are based on their in vitro and in vivo activities on glucose modulation. The current study assessed the bonding potential of isolated alkaloids with the targeted protein. For this purpose, the 3D structure of the target protein (α-glucosidase) was reproduced using MODELLER 9.20. The modeled 3D structure was then validated and confirmed by using the RAMPAGE, ERRAT, and Verify3D online servers. The molecular docking of 32 alkaloids reported as α-glucosidase inhibitors, along with reference compounds (acarbose and miglitol), was done through MOE-Dock applied in MOE software to predict the binding modes of these drug-like compounds. The results revealed that nummularine-R and vindoline possess striking interactions with active site residues of the target protein, and were analogous to reference ligands. In conclusion, the current study provided a computational background to the α-glucosidase inhibitors tested. This novel information should facilitate the development of new and effective therapeutic compounds for the treatment of diabetes mellitus.

## 1. Introduction

The digestive enzyme, α-glucosidase, has an important role in carbohydrate digestion and is responsible for the biosynthesis of glycoproteins. Several α-glucosidases can not only perform the hydrolysis of oligosaccharides and artificial α-glycosides with α-glycosidic bonds, but can also hydrolyze α-glucans such as glycogen and water-soluble starch [1,2,3]. α-glucosidase is the primary enzyme for digestion of carbohydrates in the small intestine. α-glucosidase is different from β-glucosidase because it acts on the 1,4-α bond [4,5,6,7]. For cellular growth and development in plants, glucose produced by the activity of these enzymes is used as a major energy source [8]. These enzymes are also inherent to various plant tissues, such as seeds, leaves, fruit, and roots. In the absence of α-amylase, α-glucosidase starts the breakdown of natural starch granules in different parts of plants such as barley seeds and pea chloroplasts [8,9,10]. The therapeutic potential of alkaloids has long been recognized for the treatment of various human disorders [11,12,13,14].

Alongi et al., 2018 [2] showed that conophylline **1** isolated from the leaves of *Ervatamia microphylla* exhibits significant antidiabetic effects in streptozotocin-induced diabetic rats [15]. Compounds **2**–**5** were isolated from the leaves of *Murraya koenigii* [3,4]. Compounds **6**–**8** were isolated from the stems of *Tinospora cordifolia* [16,17]. Compounds **9**–**11** were reported by Flanagan et al., 1978 [5] from the leaves of *Catharanthus roseus.* Compounds **12**–**14** were isolated from whole plant extracts of *Ziziphus oxyphylla* [18,19]. Compounds **15**–**21** were isolated from the roots and rhizomes of *Berberis lyceum*, *Coptidis rhizome*, and *Coptis japonica* [20,21,22] and compound **22** was isolated from the roots of *Berberis brevissima* and *Berberis parkeriana* [23]. Compounds **23**–**26** were reported by Arinaminpathy et al. [6] to be from the leaves of *Tecoma stans*. Compounds **27**–**31** were reported from seed extracts of *Nigella glandulifera* [7]. *Brassica oleracea* var. *capitata* seeds contain compound **32** [15,16,17,18,19,20,21,23,24,25,26].

An advantage of in silico approaches in structure-based drug design is that they minimize the time as well as the cost of developing ideas for new targets and potential lead compounds [8].

The aim of this research article was to assess the interaction of these reported antidiabetic alkaloids with target proteins such as α-glucosidase and to find novel information on active sites of α-glucosidase for the development of effective inhibitors.

## 2. Methodology

### 2.1. Target Sequence Retrieval

The protein sequence of human α-glucosidase with accession no. ABI53718.1 was downloaded from the NCBI (National Centre for Biotechnology Information) database in a FASTA format (a text-based format for representing nucleotide or peptide sequences with single-letter codes). The FASTA sequence of the protein was used for subsequent analysis to build the homology model.

### 2.2. Template Selection and Alignment

The query sequence was then used in the BLASTp program by the NCBI by selecting the Protein Data Bank (PDB) to identify homologs in the PDB (RCSB Protein Databank). We selected three templates with the PDB IDs 5KZW, 5NN4, and 5NN3 with 99% identity (https://www.rcsb.org) [9,10] for alignment and sequence identity of target protein structure prediction. Chimera 1.13 (developed by the resource for Biocomputing, Visualization, and Informatics at the University of California, San Francisco, CA, USA with support from NIH P41-GM103311) was used for target–template alignment and superposition.

### 2.3. Homology Modeling

The protein sequence was subjected to homology modeling using MODELLER 9.20 (maintained by Ben Webb at the department of Biopharmaceutical Sciences and Pharmaceutical Chemistry, and California Institute for Quantitative Biomedical Research, Mission, CA, USA). Through sequence alignment against those of previously known crystal structures, the template structure was predicted. Modeled structures per protein are generated by the MODELLER and one structure was chosen on the basis of root mean square deviation (RMSD) between the template and generated models.

### 2.4. Validation of the Modeled Structure

The obtained modeled structure of the query protein was verified for its stereo-chemical quality by using RAMPAGE (http://mordred.bioc.cam.ac.uk/~rapper/rampage.php), Verify3D (http://servicesn.mbi.ucla.edu/Verify3D/), and ERRAT (https://servicesn.mbi.ucla.edu/ERRAT/) servers to check the quality of the predicted structure.

### 2.5. Active Site Prediction

The active site was predicted by using the site finder option of using MOE (Molecular Operating Environment) software. The site finder option was used to calculate possible active sites in α-glucosidase from the 3D atomic coordinates of the receptor. Calculations were made to determine potential sites for ligand binding and docking, and restriction sets for rendering partial molecular surfaces [27].

### 2.6. Alkaloids Selection

Alkaloids are one of the most studied and widely distributed classes of secondary metabolites. The selection criteria for these alkaloids is based on their in vitro and in vivo activities on glucose inhibition. Plant alkaloids constitute 16.5% of reported natural products, and comprise almost 50% of plant-derived natural products of pharmaceutical and biological significance. About 35.9% of the reported alkaloids have been tested biologically in 20 or more assays as being pharmaceutically significant [11].

### 2.7. Preparation of Ligand for Docking Analysis

All the ligands or molecules involved in our study were collected from the available literature. All these molecular structures were reproduced in Chem-Draw ultra-version 12.0.2.1076 (2010) and then all ligands were saved in mol format with the aim to open these files in MOE after structure preparation, and these were protonate 3D at a temperature of 300 °C and pH 7 and energy minimized through MOE, using default parameters. The MMFF94× force field was used with no periodicity and the constraints were maintained at the rigid water molecule level.

### 2.8. Preparation of Protein and Molecular Docking

The modeled structure of α-glucosidase was 3D protonated and then energy minimization was performed using the MOE software with default parameters as mentioned above. For molecular docking, receptors were subjected and polar hydrogens were added. While performing docking, the ligand atom was selected and rescoring1 was set at London dG and rescoring2 at GBVI/WSA dG, running so as to note the ligand interaction with protein. Protein-ligand docking score, ligand properties, and 2D and 3D structures were saved.

## 3. Results

### 3.1. Target-Template Alignment

The target sequence of α-glucosidase that was aligned with 5KZW by the Chimera software showed 99% similarities. During alignment the target protein (α-glucosidase) was superimposed on the template sequence (5KZW) shown in Figure 1. Results of target protein and template sequence alignment are in Figure 2.

### 3.2. Homology Modeling

The homology model of the target protein, α-glucosidase, was accomplished with MODELLER 9.20. The 3D structure of the protein was modeled for further docking studies. The 3D modeled structure of the protein is shown in Figure 3.

### 3.3. Validation of the Modeled Structure

The modeled structure validated by RAMPAGE, showing stereo-chemical verification, and the verified 3D structure by ERRAT, are shown in Figure 4. For protein structure verification, a Ramachandran plot was drawn with MOE, as shown in Figure 5.

### 3.4. The Ramachandran Plot 

The plot shows:
(i)Number of residues in the favored region (~98.0% expected): 898 (94.32%)(ii)Number of residues in the allowed region (~2.0% expected): 49 (5.1%)(iii)Number of residues in the outlier region: 5 (0.52%).


### 3.5. Active Site Prediction

After sequence alignment, formation of the 3D structure and the verification of its active site, for ligands in the target protein, was predicted with the MOE software. The active site of the target protein was comprised of amino acids GLU174, THR175, AR178, GLU196, THR197, PRO198, ARG199, VAL200, HIS201, SER202, ARG203, ALA204, PRO205, GLN352, LEU355, ASP356, VAL357, VAL358, GLY359, TYR360, ARG608, VAL718, ALA719.

### 3.6. Preparation of Protein and Molecular Docking

Proteins were prepared for molecular docking by 3D protonation, energy minimization and prediction of active site for ligands, keeping the parameters at their defaults. Next, ligands were docked with the target protein (α-glucosidase) while using MOE software. The docking results suggested that Nummularine-R was the most potent of the tested compounds, with a docking score of −14.5691 followed by Vindoline with a docking score of −13.2250. In addition to these two compounds, Conophyline, Epiberberine, glutamic acid, and mahanimbilylacetate also showed favorable results. The docking results, along with ligand structure and their properties, are shown in Table 1.

## 4. Discussion

Molecular docking is frequently used to predict the binding orientation of small molecules and drug candidates to their protein targets in order to predict their affinity and activity [28,29]. In this study, we modeled various alkaloids isolated from different plants and known for their inhibition of α-glucosidase through molecular docking.

Glucose control is an effective and long-lasting treatment for type II diabetes mellitus, minimizing both cardio-vascular and nervous system symptoms associated with the disease [30,31]. α-glucosidase inhibitors are usually recommended for diabetic patients to decrease postprandial hyperglycemia caused by the breakdown of starch molecules in the small intestine [32]. The use of plants or plant-based substances may be a suitable source of α-glucosidase inhibitors because of their low price and comparatively greater safety, with a low frequency of serious gastrointestinal side effects [33].

The docked alkaloids also exhibit inhibitory potential against other hydrolase enzymes in the same class. Previous reports [21] shows that alkaloids such as berberine and palmatine inhibit aldose reductase activity at lower concentrations, losing their inhibitory potential at higher concentrations. On the other hand, coptisine, epiberberine, and groenlandicine showed reasonable inhibitory potential. Nigelladines A–C, pyrroloquinoline and nigellaquinomine have shown strong protein tyrosine phosphatase inhibitory activity [25].

Among the 10 differencing docking alkaloids, nummularine-R was found to be the most potent of the tested compounds with a docking score of −14.5691, followed by Vindoline with a docking score of −13.2250. Both these compounds have good inhibitory activity and their docking score is in the region of certain standard ligands, such as miglitol (−15.4423) and acarbose (−14.7983). Furthermore, both these ligands exhibited a good interaction with α-glucosidase. The most potent ligand, nummularine-R, formed four hydrogen interactions with the Gln121, Met122, Arg331, and Gly546 active amino acid residues. Gln121 was observed to make a polar hydrogen bond, with the oxygen atom double bonding with the piperidine moiety of the ligand. Arg331 formed polar hydrogen bonds with the Nitrogen atom of the pentene ring of the ligand. Met122 showed acidic hydrogen interactions, whereas Gly546 showed basic hydrogen interactions with the oxygen atom double bonding with the piperidine moiety of the same ligand. These interactions are shown in Figure 6A. Ala93, Ala97, Gln121, and Trp126 formed three Hydrogen and one arene-arene interaction with the ligand. Ala93 showed greasy hydrogen interaction with the nitrogen of the benzene ring, with a bond length of 3.07 Å, while Gln121 showed greasy hydrogen interaction with the hydrogen of the benzene ring, with a bond length of 3.01 Å. Ala97 demonstrated a basic hydrogen bond with an oxygen atom double bonding with the piperidine moiety of the ligand with a bond length of 3.07 Å. Trp126 exhibited an arene-arene bond with a bond length of 3.89 Å with the benzene ring of the inhibitor. All these interactions are shown in Figure 6B.

The reference compound Miglitol revealed four hydrogen bonds with Met122, Arg275 and Arg331 of the target protein active site residue. Met122 formed two acidic hydrogen bonds with the hydrogen and oxygen of the same phenol moiety, while Arg331 bound via polar hydrogen interaction with the oxygen of the benzene ring of the phenol moiety of the ligand. Arg275 formed a polar hydrogen bond with the oxygen atom, double bonding with the piperidine moiety of the same compound. All these bonds are shown in Figure 6C. Another reference compound, Acarbose, showed seven hydrogens, Ala93, Ile98, Gln121, Met122, Arg275, Pro545 and one arene-cation interaction with Trp126 of the target protein. Ala93 and Met122 formed greasy and acidic hydrogen bonds with the nitrogen of the benzene ring. Met122 also formed acidic hydrogen with the carbon backbone of the benzene ring. Gln121 formed a greasy hydrogen interaction with the carbon of the benzene backbone. Arg275 exhibited a polar hydrogen bond with the OH group of the phenol moiety, similarly Pro545 possessed greasy hydrogen and Ile98 formed basic hydrogen interactions with the target protein. Trp126 exhibited an arene-cation bound with the carbon backbone of the ligand. All these bonds are shown in Figure 6D. The 3D interaction of the most potent ligands (nummularine-R and Vindoline) and standard (Acarbose and Miglitol) with target protein are shown in Figure 7, Figure 8, Figure 9 and Figure 10, respectively. Similarly, these docking results were consistent and in full agreement with the in vitro anti-diabetic activity previously reported [18,19]. In addition to nummularine-R and Vindoline, Conophyline, Epiberberine, Glutamic acid and Mahanimbilylacetate also showed good interactions with the target protein, with docking scores of −12.6274, −12.9822, −12.6023, −12.9971, −12.7703, respectively.

Nummularine-R, formed four hydrogen interactions, with the Gln121, Met122, Arg331, and Gly546 active amino acid residues.

## 5. Conclusions

The molecular docking of 32 alkaloids isolated from various plants, along with the standard compounds acarbose and miglitol, were docked to α-glucosidase by using MOE-Dock applied in MOE software to predict the binding modes of these drug-like compounds. The results showed that nummularine-R and Vindoline possessed striking interactions with active site residues of the target protein, α-glucosidase, and were analogous to reference ligands. Taken together, the current study provides a computational background for several α-glucosidase inhibitors. Future studies should more carefully examine the clinical efficacy of these compounds, thus facilitating the development of novel resources for the treatment of diabetes mellitus.

## Figures and Tables

**Figure 1 biomolecules-09-00544-f001:**
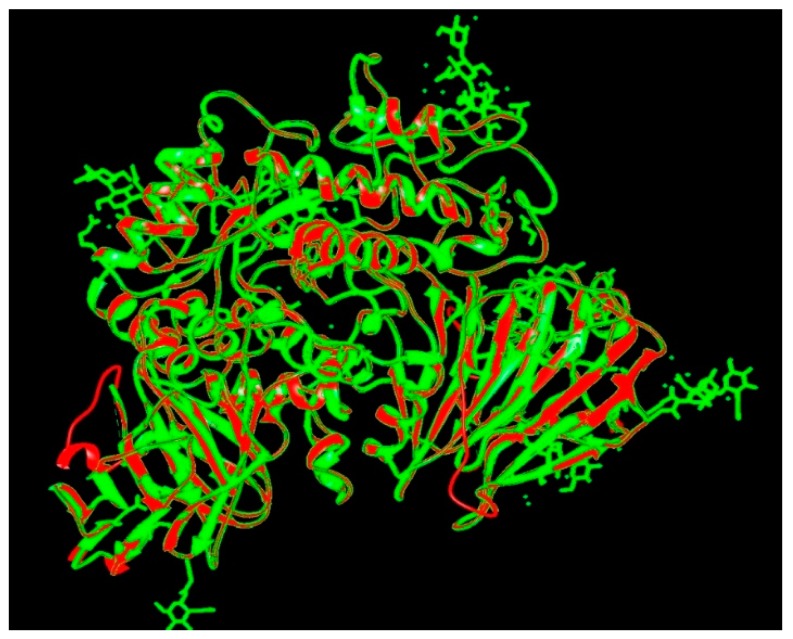
Superposition of target (red) and template (green).

**Figure 2 biomolecules-09-00544-f002:**
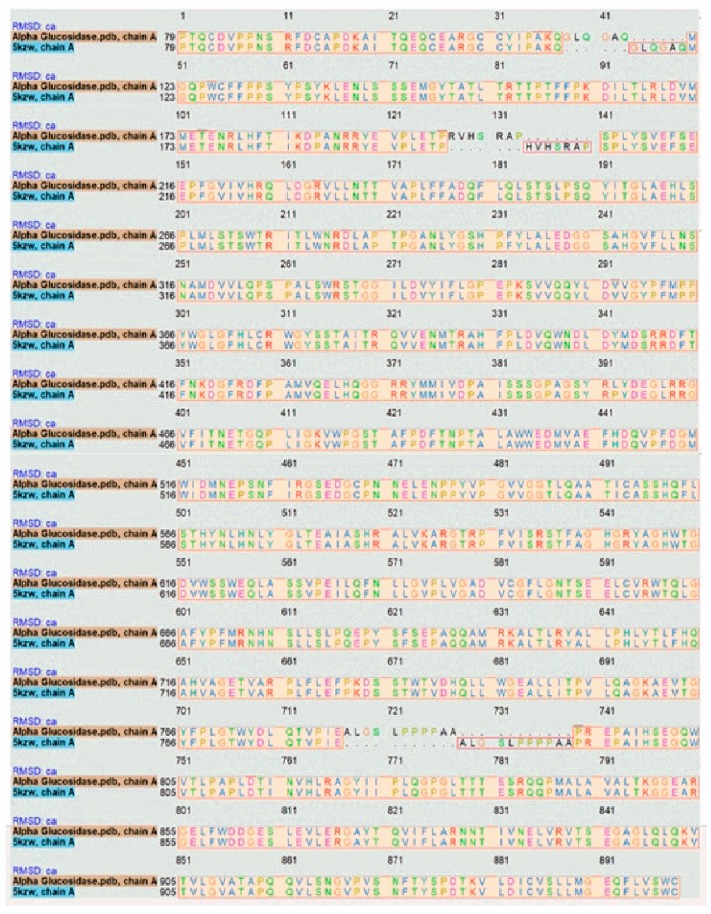
Alignment of target protein with the template.

**Figure 3 biomolecules-09-00544-f003:**
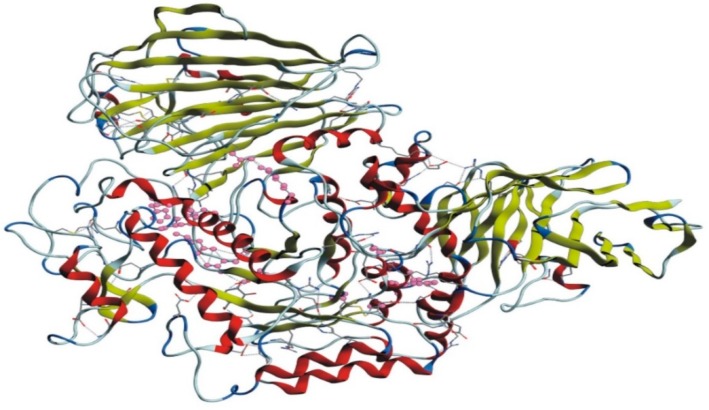
3D modeled structure of α-glucosidase.

**Figure 4 biomolecules-09-00544-f004:**
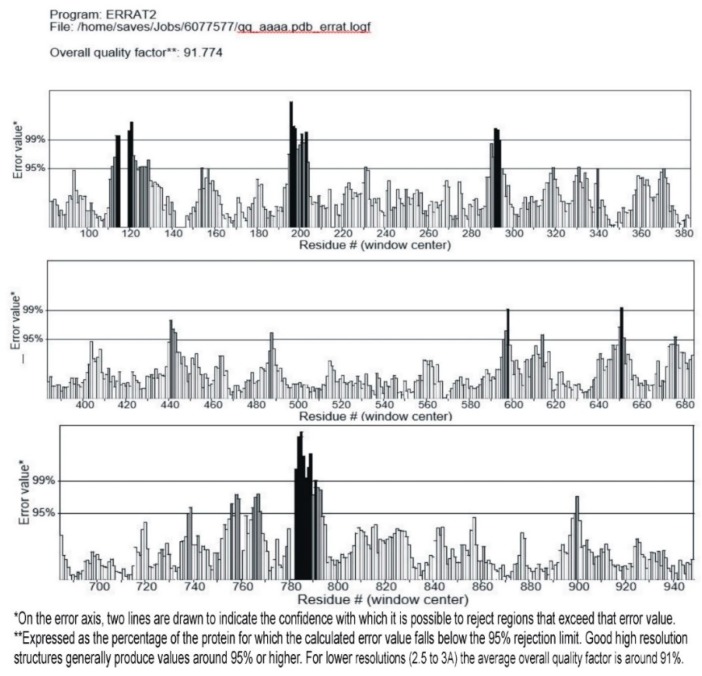
Showing the 95% of amino acids in ideal range and are below rejection level.

**Figure 5 biomolecules-09-00544-f005:**
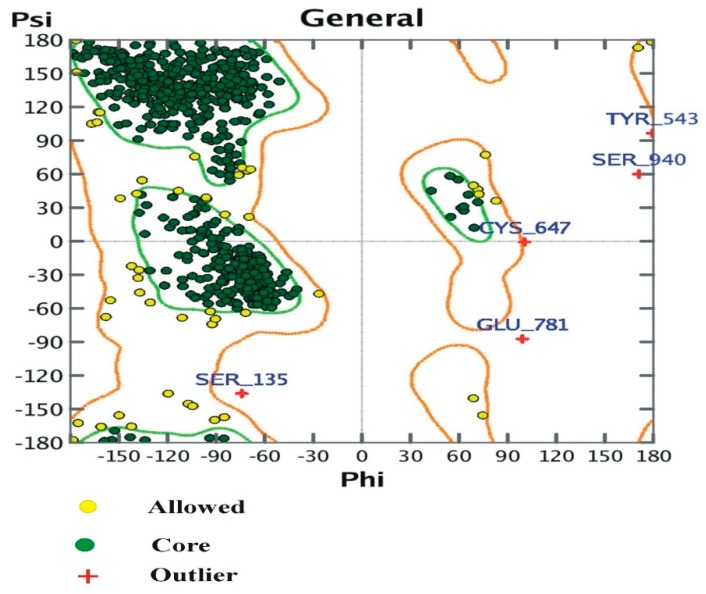
Ramachandran plot and position of amino acids in α-glucosidase.

**Figure 6 biomolecules-09-00544-f006:**
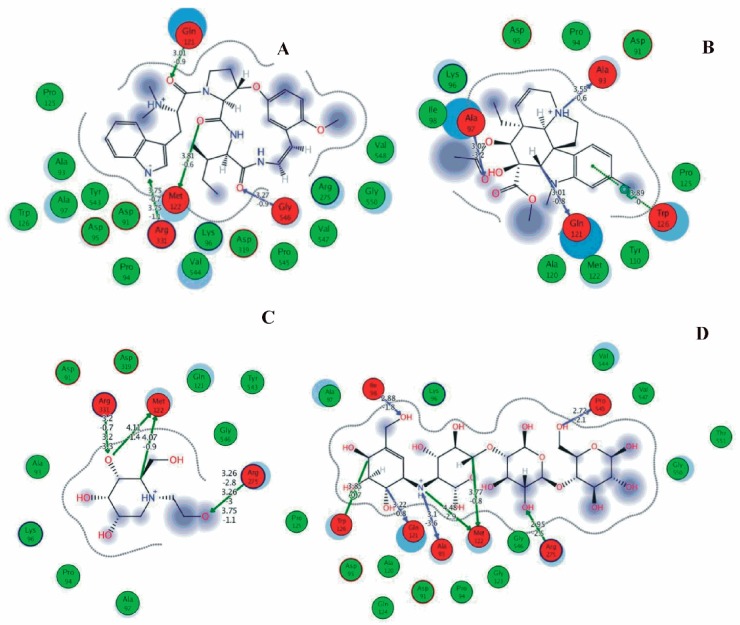
2D images of the docked conformations of the ligands and standard with the active residues (**A**) 2D image of nummularine-R (**B**) 2D image of Vindoline (**C**) 2D image of miglitol, and (**D**) 2D image of acarbose.

**Figure 7 biomolecules-09-00544-f007:**
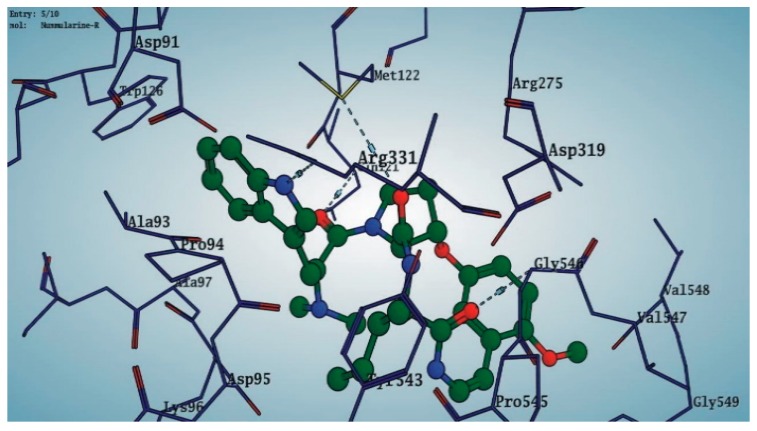
3D interaction of most active compound **14** (Nummularine-R).

**Figure 8 biomolecules-09-00544-f008:**
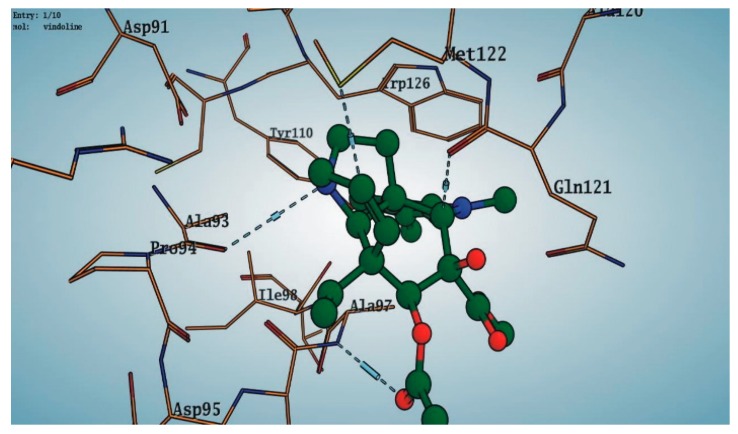
Interaction of compound **10** (Vindoline) with Ligand target protein. Ala93, Ala97, Gln121, and Trp126 form three Hydrogen and one arene-arene interaction with the ligand.

**Figure 9 biomolecules-09-00544-f009:**
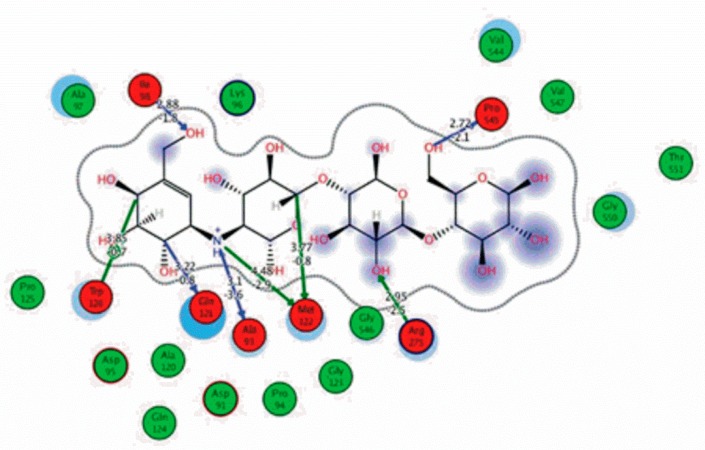
2D interactions of standard (acarbose). Acarbose showed seven hydrogen, Ala93, Ile98, Gln121, Met122, Arg275, Pro545, and one arene-cation interaction with Trp126 of the target protein.

**Figure 10 biomolecules-09-00544-f010:**
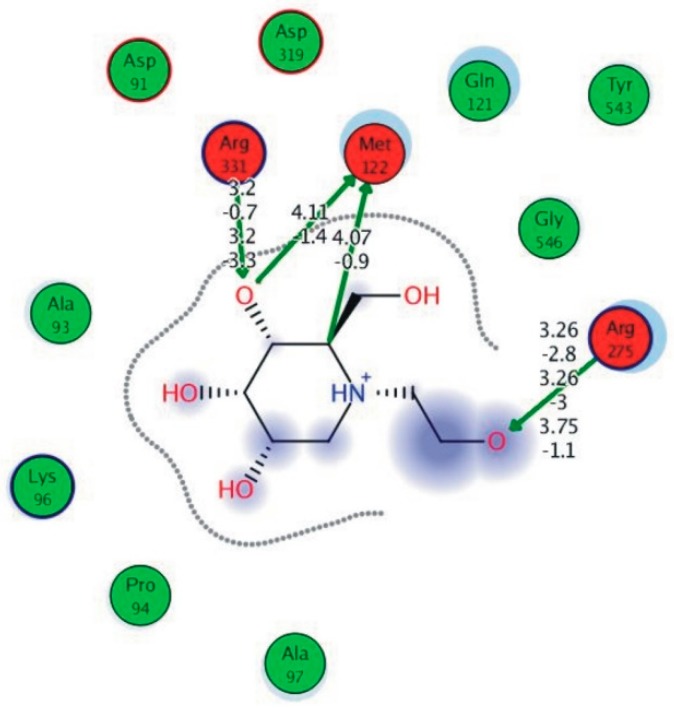
Interaction of reference compound (Miglitol). Miglitol shows four hydrogen bonds with Met122, Arg275, and Arg331 of the target protein active site residue.

**Table 1 biomolecules-09-00544-t001:** Ligand chemical structures, their properties, and docking scores.

Plant	Molecular Structures	Activity	Docking Score	Ref.
*Ervatamia microphylla*(leaves)	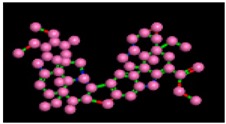 Conophyline **1**	In vivo p.o. stimulate iPMSCs proliferation	−12.6274	[1]
*Murraya koenigii*(leaves)	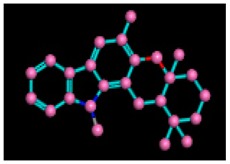 Bicyclomahanimbiline **2**	In vivo p.o.HypoglycemicActivity	−11.7634	[2]
*Murraya koenigii*(leaves)	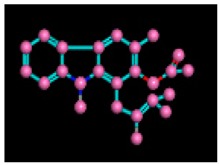 Girinimbilylacetat **3**	In vivo p.o.HypoglycemicActivity	−9.5231	[2]
*Murraya koenigii*(leaves)	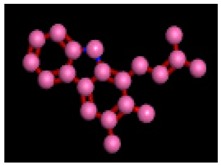 Girinimbine **4**	In vivo p.o.HypoglycemicActivity	−9.9589	[2]
*Murraya koenigii*(leaves)	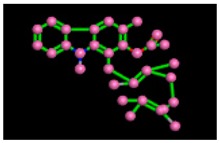 Mahanimbilylacetate **5**	In vivo p.o.HypoglycemicActivity	−12.9971	[2]
*Coptis chinensis*(Rhizome)	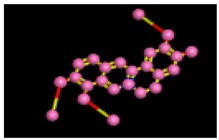 Jatrorrhizine **6**	In vitro anti-diabetic	−9.3385	[3,4]
*Coptis chinensis*(Rhizome)	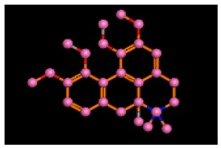 Magnoflorine **7**	In vitro anti-diabetic	−11.2586	[3,4]
*Coptis chinensis*(Rhizome)	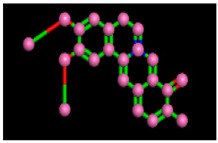 Palmatine **8**	In vitro Anti-diabetic	−10.0536	[3,4]
*Catharanthus roseus* (Leaves)	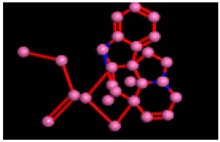 Vindolicine **9**	In vitro Anti-diabetic	−9.2272	[5]
*Catharanthus roseus* (Leaves)	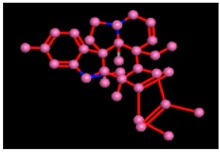 Vindoline **10**	In vitro Anti-diabetic	−13.2250	[5]
*Catharanthus roseus* (Leaves)	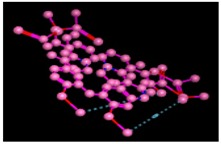 Vindolinine **11**	In vitro Anti-diabetic	−5.5275	[5]
*Ziziphus oxyphylla*(Whole plant)	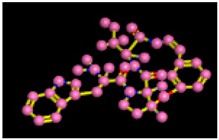 Hemsine-A **12**	In vitro Control the postprandial hyperglycemia	−10.4509	[6]
*Ziziphus oxyphylla*(Whole plant)	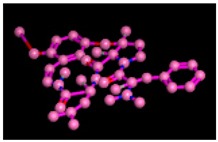 Nummularin-C **13**	In vitro Anti-diabetic Control the postprandial hyperglycemia	−10.3726	[6]
*Ziziphus oxyphylla*(Whole plant)	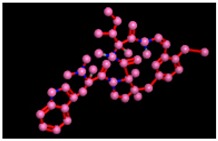 Nummularine-R **14**	In vitro Anti-diabetic Control the postprandial hyperglycemia	−14.5691	[6]
*Berberis lyceum* (Root)	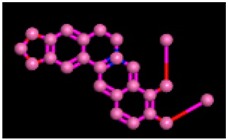 Berberine **15**	In vitro Anti-diabetic Hypoglycemic Activity	−10.5667	[7]
*Coptis japonica* (Root)	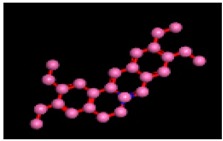 Columbamine **16**	In vitro Anti-diabetic Aldose Reductase Inhibitory Activity	−7.4609	[3]
*Coptis chinensis* (*Rhizome*)	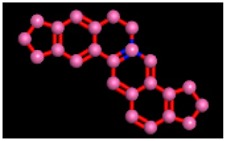 Coptisine **17**	Anti-diabetic	−8.9123	[3,4]
*Coptis chinensis* (*Rhizome*)	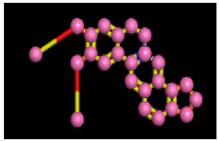 Epiberberine **18**	In vitro Anti-diabetic	−12.9822	[3,4]
*Coptis japonica* (Root)	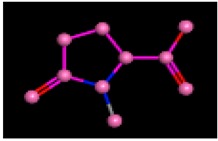 Glutamic acid **19**	In vitro Anti-diabetic Aldose Reductase Inhibitory Activity	−12.6023	[3]
*Coptis chinensis* (*Rhizome*)	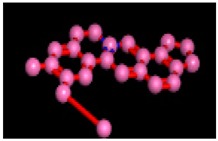 Groenlandicine **20**	In vitro Anti-diabetic	−7.0817	[3,4]
*Coptis chinensis* (*Rhizome*)	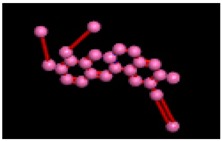 Jateorrhizine **21**	In vitro Anti-diabetic	−11.4544	[3,4]
*Coptis japonica* (Root)	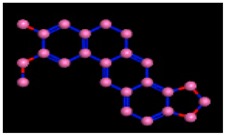 Dehydrocheilanthifoline **22**	In vitro Anti-diabetic Aldose Reductase Inhibitory Activity	−10.8606	[3]
*Tecoma stans* (Leaves)	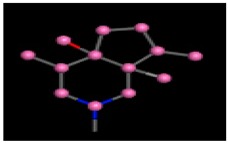 β-hydroxyskitanthine **23**	In Vivo andIn Vitro Potent stimulating effect on the basal glucoseuptake rate	−10.2216	[8]
*Tecoma stans* (Leaves)	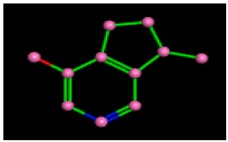 Boschnlakine **24**	In Vivo andIn Vitro Potent stimulating effect on the basal glucoseuptake rate	−7.6929	[8]
*Tecoma stan* (Leaves)	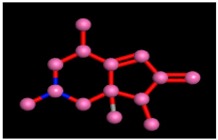 Tecomine **25**	In Vivo andIn Vitro Potent stimulating effect on the basal glucoseuptake rate	−9.1085	[8]
*Tecoma stans* (Leaves)	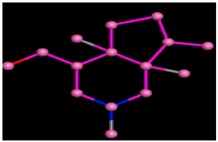 Tecostanine **26**	In Vivo andIn Vitro Potent stimulating effect on the basal glucoseuptake rate	−9.9845	[8]
*Nigella**glandulifera.* (Seed)	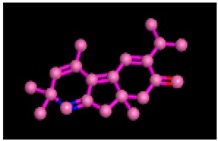 Nigelladines A **27**	In Vitro PTP1B inhibitory activity	-	[9]
*Nigella**glandulifera.* (Seed)	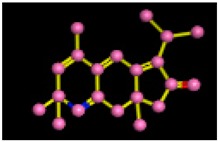 Nigelladines B **28**	In Vitro PTP1B inhibitory activity	−9.7263	[9]
*Nigella**glandulifera.* (Seed)	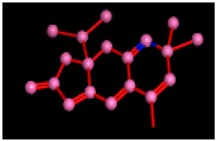 Nigelladines C **29**	In Vivo andIn Vitro PTP1B inhibitory activity	−9.9462	[9]
*Nigella**glandulifera.* (Seed)	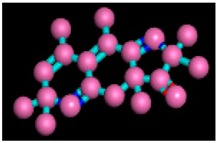 Nigellaquinomine **30**	In Vitro PTP1B inhibitory activity	−10.7638	[9]
*Nigella**glandulifera.* (Seed)	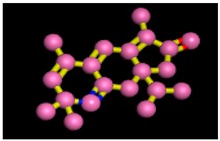 Pyrroloquinoline **31**	In Vitro PTP1B inhibitory activity	−9.4846	[9]
*Brassica oleracea* var. capitate (Seed)	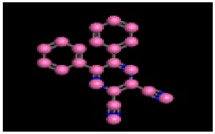 2,3-Dicyano-5,6-diphenylpyrazine **32**	Antidiabetic activity	−9.6067	[10]
**34**	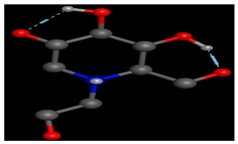 Miglitol		−15.4423	
**35**	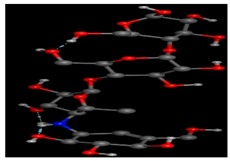 Acarbose		−14.7983

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
