# Peer review of "Molecular Docking of Isolated Alkaloids for Possible α-Glucosidase Inhibition"

_biomolecules, 2019, doi:10.3390/biom9100544_

Round 1

Reviewer 1 Report

α-Glucosidase inhibitory potential of isolated alkaloids through Molecular docking 

Please correct in the paper:

.    Same way of writing α-Glucosidase

.    Line 46 – reference

.    Line 47, 48 – reference

.    Line 95 – bold

.    Line 152 – font

.    Referenca number correct 1, 5, 7, 8, 9, 10, 11, 13, 14, 18, 19, 20, 22, 28, 29, 32, 33

.    Line 187 – please explain why you didn't show results for Quercitrin in table

.     The resolution of the Figure 2 and 4 should be improved

Author Response

Dear Editor

Many thanks for the review of our article. We have made every effort to address all point/corrections suggested by the reviewers.  I am sure the incorporation of these suggestions will greatly add to the overall quality of our article. 

1st Reviewer

Comments and Suggestions for Authors

α-Glucosidase inhibitory potential of isolated alkaloids through Molecular docking.

Please correct in the paper:

.    Same way of writing α-Glucosidase
  ……

Reply: Corrected accordingly throughout the MS

_______________________________________________________________________________________

.    Line 46 – reference…….

Reply: Reference has been added.

___________________________________________________________________________________

.    Line 47, 48 – reference….

Reply: needful is done

__________________________________________________________________________________

. Line 95 – bold……………

Reply: Changed accordingly

_______________________________________________________________________________

.    Line 152 – font…………

Reply: Changed accordingly

.    Reference number correct 1, 5, 7, 8, 9, 10, 11, 13, 14, 18, 19, 20, 22, 28, 29, 32, 33

Reply: Formatted according to the Journal style

_____________________________________________________________________________

.    Line 187 – please explain why you didn't show results for Quercitrin in table?

Reply: We are thankful to you for pointing out this compound. It was mistakenly added as an alkaloid, but it is flavonoid, not an alkaloid.

__________________________________________________________________________________.     

The resolution of the Figure 2 and 4 should be improved

Reply: The best possible resolution is added to the MS.

__________________________________________________________________________________

Thanks

Reviewer 2 Report

The manuscript is very well written and finding alkaloid based compounds inhibitors of alpha-glucosidase is of interest as alternative approaches for treatment of diabetes mellitus.  

Please add a comment in the results and discussion about the selectivity of the tested alkaloids for other type of enzymes from the same hydrosylase class.

Author Response

2nd Reviewer

Many thanks for the valuable comments and suggestions and I am sure it will greatly add to the overall strength of our article.

Comments and Suggestions for Authors

The manuscript is very well written and finding alkaloid based compounds inhibitors of alpha-glucosidase is of interest as alternative approaches for treatment of diabetes mellitus.  

Please add a comment in the results and discussion about the selectivity of the tested alkaloids for other type of enzymes from the same hydrosylase class.

Reply: As per the available literature, we have added a paragraph on page 13, line 191-197 about the tested alkaloids for other hydrolyase enzymes.

_______________________________________________________________________

Reviewer 3 Report

The manuscript entitled "alpha-Glucosidase inhibitory potential of isolated alkaloids through Molecular docking" reports a study about selecting alkaloids as potential inhibitors of α-glucosidase that is important for the digestion of the carbohydrates. For this purpose, the authors used several in silico methods to perform a structure-based analysis. The manuscript can be accepted, however, there are some points to be changed. The main point is the choice of the compounds used in this study, what are the criteria used?

1 - Abstract - page 1 - lines 22-24
It is not clear the criteria of choice of the alkaloids. Please, clarify succinctly in the abstract.

2 - Tittle - page 1
It is suitable to change the title, one reading the title realizes wrongly that the authors isolate the compounds.

3 – Introduction – page 2
It is not clear the criteria of the choice of the compounds 2-32 used in this study. Please rewrite the paragraph.

4 - Introduction – page 2 – line 46
“Dineshkumar et al 2010 isolated compounds 2-5 from the leaves of Murraya koenigii. [reference].”
Please add the citation “[reference]”.

5 – Introduction – page 1 – line 56
“such as α-glucosidase, with the imparting novel information”
Please, standardize the font size.

6 – Introduction
Please add a sentence explaining the advantages of the use of the in-silico approaches in the structure-based drug design.

7 – Methodology
Please, add a subsection dataset, explaining the selection of the compounds.

8 – Methodology – page 2 lines 84-85
It is suitable to add more details of the preparation of the ligands, mainly regarding “were protonate 3D and energy minimized through MOE using default parameters”, for instance, what kind of force field minimization was used? Some inspection of the compounds of pre-treatment/standardization regarding the structures, please clarify.

9 – Methodology – page 2 - Template Selection and Alignment
“Templates with PDB ID`s 5KZW”
Please, clarify, only one template was selected, the 5KZW?
Please, cite the PDB ID 5KZW:
10.2210/pdb5KZW/pdb

10 – Methodology - page 3 – “Preparation of Protein and Molecular Docking”
Please, clarify the docking process. How the criteria of the active site/pockets used? Clarify the in methods used, not only citing the software used. It is interesting, strongly recommended to add some result of the validation of the prediction of the active site.

11 - Results and discussion
Please, the authors must improve the discussion regarding some critical residues. I think that a table of some critical residues and king of interactions comparing nummularine-R, vindoline, acarbose and miglitol is more useful than figures.

12 – Results and discussion
I think that summarize the important structural features for an alkaloid for binding the α-glucosidase is important for one that will read the manuscript for further studies.

13 – Results and discussion
Perhaps the manuscript will increase in information if the authors searching some similar structures (alkaloids) of nummularine-R and vindoline and perform the docking. These alkaloids could be secondary metabolites or from other sources. Some examples: erantinine e , macrocarpine a, 6-oxoleuconoxine, arboloscine, ac1l4150, ac1l7ka1.

14 – Table 1 – page 7
Please, replace the figures 3D for figures 2D and add the structures of miglitol, acarbose.

15 – Discussion - Page 12 – line 148
“Nummularine-R”, please replace the names of the compounds in lower case.

Author Response

3rd Reviewer

We have made every effort to address all point/corrections suggested.  I am sure the incorporation of these suggestions will greatly add to the overall quality of our article.

Comments and Suggestions for Authors

The manuscript entitled "alpha-Glucosidase inhibitory potential of isolated alkaloids through Molecular docking" reports a study about selecting alkaloids as potential inhibitors of α-glucosidase that is important for the digestion of the carbohydrates. For this purpose, the authors used several in silico methods to perform a structure-based analysis. The manuscript can be accepted, however, there are some points to be changed. The main point is the choice of the compounds used in this study, what are the criteria used?

1 - Abstract - page 1 - lines 22-24
It is not clear the criteria of choice of the alkaloids. Please, clarify succinctly in the abstract.

Reply: we have added a sentence in the abstract and also add a small paragraph about the criteria of alkaloids selection.

___________________________________________________________________________________

2 - Tittle - page 1
It is suitable to change the title, one reading the title realizes wrongly that the authors isolate the compounds.

Reply: needful modification has been made in the revised MS

_________________________________________________________________________________

3 – Introduction – page 2
It is not clear the criteria of the choice of the compounds 2-32 used in this study. Please rewrite the paragraph.

Reply: We have added a short paragraph about the selection criteria of the alkaloids and their In-vitro and In-vivo activities are mentioned in table 1.

____________________________________________________________________________________

4 - Introduction – page 2 – line 46
“Dineshkumar et al 2010 isolated compounds 2-5 from the leaves of Murraya koenigii. [reference].”
Please add the citation “[reference]”.

Reply: Citation inserted where suggested.

____________________________________________________________________________________

5 – Introduction – page 1 – line 56
“such as α-glucosidase, with the imparting novel information”
Please, standardize the font size.

Reply: Needful is done

___________________________________________________________________________________

6 – Introduction
Please add a sentence explaining the advantages of the use of the in-silico approaches in the structure-based drug design.

Reply: As per suggestion, added a sentence at the end of the Introduction and highlighted

________________________________________________________________________________

7 – Methodology
Please, add a subsection dataset, explaining the selection of the compounds.

Reply: Added a small paragraph about the criteria for alkaloids selection and highlighted.

_____________________________________________________________________________________

8 – Methodology – page 2 lines 84-85
It is suitable to add more details of the preparation of the ligands, mainly regarding “were protonate 3D and energy minimized through MOE using default parameters”, for instance, what kind of force field minimization was used? Some inspection of the compounds of pre-treatment/standardization regarding the structures, please clarify.

Reply: Added the details for Molecular docking and highlighted.

_____________________________________________________________________________________

9 – Methodology – page 2 - Template Selection and Alignment
“Templates with PDB ID`s 5KZW” Please, clarify, only one template was selected, the 5KZW?
Please, cite the PDB ID 5KZW: 10.2210/pdb5KZW/pdb

Reply: We built the homology model through Modeller and select three templates with with PDB ID`s 5KZW, 5NN4, 5NN3 with all 99% identity for alignment. The malign3d command in Modeller performs an iterative least-squares superposition of the 3D structures, using the multiple sequence alignment.

Cited the structures

_______________________________________________________________________

10 – Methodology - page 3 – “Preparation of Protein and Molecular Docking”
Please, clarify the docking process. How the criteria of the active site/pockets used? Clarify the in methods used, not only citing the software used. It is interesting, strongly recommended to add some result of the validation of the prediction of the active site.

Reply: The required information has been added the details in the methodology section.

__________________________________________________________________________________

11 - Results and discussion
Please, the authors must improve the discussion regarding some critical residues. I think that a table of some critical residues and kind of interactions comparing nummularine-R, vindoline, acarbose and miglitol is more useful than figures.

Reply: We included the important or active site residues in the methodology section page number 7, line 166-169. In the results and discussion section we mentioned the active site residues interaction and the figures are self-explanatory.

_____________________________________________________________________________________

12 – Results and discussion
I think that summarize the important structural features for an alkaloid for binding the α-glucosidase is important for one that will read the manuscript for further studies.

Reply: the selected alkaloids have antidiabetic properties use different mechanisms by targeting different enzymes to regulate the carbohydrate metabolism. However, reports on the structural activity relationship of the alkaloids with antidiabetic properties are scarce in the literature for these alkaloids.

_________________________________________________________________________________

13 – Results and discussion
Perhaps the manuscript will increase in information if the authors searching some similar structures (alkaloids) of nummularine-R and vindoline and perform the docking. These alkaloids could be secondary metabolites or from other sources. Some examples: erantinine e , macrocarpine a, 6-oxoleuconoxine, arboloscine, ac1l4150, ac1l7ka1.

Reply: we searched the literature for the suggested alkaloids and some others but they don`t have in-vitro or in-vivo activities and we selected alkaloids having in-vitro or in-vivo anti-diabetic activity. 

_____________________________________________________________________________________

14 – Table 1 – page 7
Please, replace the figures 3D for figures 2D and add the structures of miglitol, acarbose.

Reply: Replaced the structures and also added the structures of standards i-e Miglitol and Acarbose

_____________________________________________________________________________________

15 – Discussion - Page 12 – line 148
“Nummularine-R”, please replace the names of the compounds in lower case.

Reply: Replaced with lower case

__________________________________________________________________________________

Prof. Dr. Haroon Khan